# Identification of Epigenetic Biomarkers of Adolescent Idiopathic Scoliosis Progression: A Workflow to Assess Local Gene Expression

**DOI:** 10.3390/ijms25105329

**Published:** 2024-05-14

**Authors:** Simona Neri, Elisa Assirelli, Marco Manzetti, Giovanni Viroli, Marco Ialuna, Matteo Traversari, Jacopo Ciaffi, Francesco Ursini, Cesare Faldini, Alberto Ruffilli

**Affiliations:** 1Medicine and Rheumatology Unit, IRCCS Istituto Ortopedico Rizzoli, 40136 Bologna, Italy; elisa.assirelli@ior.it (E.A.); jacopo.ciaffi@ior.it (J.C.); francesco.ursini@ior.it (F.U.); 21st Orthopaedic and Traumatologic Clinic, IRCCS Istituto Ortopedico Rizzoli, 40136 Bologna, Italy; marco.manzetti@ior.it (M.M.); giovanni.viroli@ior.it (G.V.); marco.ialuna@ior.it (M.I.); matteo.traversari@ior.it (M.T.); cesare.faldini@ior.it (C.F.); alberto.ruffilli@ior.it (A.R.); 3Department of Biomedical and Neuromotor Sciences (DIBINEM), Alma Mater Studiorum University of Bologna, 40126 Bologna, Italy

**Keywords:** adolescent idiopathic scoliosis, epigenetics, gene expression, DNA methylation, spinal facet, paravertebral muscle, spinal ligament

## Abstract

Adolescent idiopathic scoliosis (AIS) is a three-dimensional structural deformity of the spine that affects 2–3% of adolescents under the age of 16. AIS etiopathogenesis is not completely understood; however, the disease phenotype is correlated to multiple genetic loci and results from genetic–environmental interactions. One of the primary, still unresolved issues is the implementation of reliable diagnostic and prognostic markers. For clinical management improvement, predictors of curve progression are particularly needed. Recently, an epigenetic contribution to AIS development and progression was proposed; nevertheless, validation of data obtained in peripheral tissues and identification of the specific mechanisms and genes under epigenetic control remain limited. In this study, we propose a methodological approach for the identification of epigenetic markers of AIS progression through an original workflow based on the preliminary characterization of local expression of candidate genes in tissues directly involved in the pathology. The feasibility of the proposed methodological protocol has been originally tested here in terms of identification of the putative epigenetic markers of AIS progression, collection of the different tissues, retrieval of an appropriate amount and quality of RNA and DNA, and identification of suitable reference genes.

## 1. Introduction

Adolescent idiopathic scoliosis (AIS) is a three-dimensional structural deformity of the spine in healthy growing adolescents, involving both frontal, sagittal, and axial planes. It is defined by a greater than 10° lateral deviation of the spine in the frontal plane, in the absence of vertebral anomalies and without association to any specific disease [1,2]. AIS is diagnosed by exclusion, made when all other potential causes of scoliosis, including vertebral malformation, neuromuscular disease, and syndromic disorders, have been ruled out. According to studies, AIS prevalence ranges between 0.35% and 5.2%, with an average of 2–3% in adolescents under the age of 16 [3].

Scoliotic curves progress until skeletal maturity, generating significant aesthetic concerns such as humps, as well as psychological issues like low self-esteem, coronal and/or sagittal imbalance, and muscular fatigue. In rare situations, curve progression can result in a severe deformity with lung restrictive illness, an increase in right atrial and ventricular pressures, and neurological damage [2].

Despite extensive clinical, epidemiological, and basic scientific research, the etiopathogenesis of AIS remains unclear; in fact, this complex phenotype results from several factors including neuromuscular dysfunction [4], abnormal skeletal maturation, environment [5], and the interaction of multiple genetic loci with each other and the environment [6].

Early AIS diagnosis and prognosis will potentially improve treatments, but biomarker identification is currently a challenge. While the Cobb angle is considered effective for initial curve severity assessment and is generally performed as the first-step evaluation tool, physicians require other predictors to define the curve progression risk at the time of diagnosis. Identifying predictors of curve progression is essential to prevent inadequate clinical management that denies patients access to proper treatment or exposes others to unnecessary ones.

Indeed, after a diagnosis is made, patients require different treatment strategies (from observation alone, to orthotic treatment and surgical correction) in accordance with curve magnitude at the time of diagnosis and to curve progression potential. Surgical treatment is currently the definitive treatment for patients with severe curve or with conservative treatment failure, achieving powerful curve correction, but correlated with high morbidity and intra- or post-operative complications [2].

While numerous clinical characteristics (such as major curve localization and Cobb angle, Risser and Tanner stage, and triradiate cartilage status) [7,8] are commonly regarded as predictors, little is known about genetic and epigenetic factors.

Genetic studies have identified several gene variants/polymorphisms associated with AIS progression, including, among others, *ER*, *IGF-1*, *TPH-1*, *LBX1*, and *FBN1* loci [9]. Unfortunately, these associations showed low predictive capacity, and, to date, no single genetic marker or combination of genetic markers has been able to produce a valid genetic model for predicting the course of the disease. Studies on epigenetic markers of AIS progression are more recent and, despite the limited scientific evidence, results are strongly encouraging [9,10]. However, no specific markers were identified, and most of the available data were obtained on DNA from peripheral blood [11]. Conversely, few results were obtained on the musculoskeletal tissues directly affected [12], and very few compared these tissues with each other [13]. A general association of disease progression with demethylation was observed in some studies [14,15]. Some genes undergoing epigenetic regulation in association with disease progression emerged as putative biomarkers, some demethylated during AIS progression (*HAS2* [16], *WNT* genes [14], *NPY* [14]), and some methylated (*COMP*, *PITX1*, and *PCDH10*) [17,18]. Finally, some miRNAs showed correlation between their expression and disease severity [19,20]. Recently, circulating extracellular vesicles from severe AIS females demonstrated a peculiar expression of members of the miR-30 family compared to controls. These vesicles were able to affect in vitro osteogenic differentiation of mesenchymal stromal cells, thus suggesting a contribution to disease pathogenesis and severity [21].

The aim of the present study is to provide an original protocol for a proper approach to the identification of epigenetic markers of AIS progression. The very interesting data emerged from large-scale studies carried out on peripheral blood, and highlighting alterations in the methylation status of several DNA regions first requires validation in the tissues directly involved in the disease, as a prerequisite for identifying the mechanisms of disease progression and developing reliable predictive biomarkers.

### Proposed Investigation Protocol

In this manuscript, we propose an investigation protocol useful for identifying and testing putative epigenetic regulators of a biologic mechanism, here applied to AIS. The principal steps of the proposed workflow are shown in Figure 1 and described in detail below. They include the following: identification of the promising epigenetic biomarkers from available literature data; recruitment of donors and associated clinical data; collection of surgical samples from donors’ paravertebral muscle, bone, and ligament together with peripheral blood; DNA and RNA extraction from samples; analysis of gene expression of the putative biomarkers in the different tissues followed by study of the epigenetic mechanisms of regulation of gene expression in the DNA of the same samples. In the following sections, the hypothesis made for the implementation of the steps of the proposed protocol is described.


*Step 1: Literature search and gene panel identification*


Knowing the specific epigenetic biomarkers influencing the curve progression of AIS patients during growth is essential for identifying reference input data (type of biomarker, analyzed tissues, and analysis techniques). To define a specific panel of putative genes of interest, the first step involves a relevant literature search of available knowledge concerning AIS curve progression and putative epigenetic regulated genes. A literature data search is essential to determining which epigenetic regulators to study and how to proceed with their collection and analysis.


*Step 2: Sample and data collection*


In the second step, the criteria to enroll patients, the data to be collected from patients, and the tissues to be analyzed are defined.

Study groups and control groups are defined based on the experimental plan and the specific research goals. Comparison between patients and controls and between patients at different stages of the disease or among different tissues of the same patient can be planned.

Collection of biological specimens takes place during the surgical procedure. Spine surgery relies on placing pedicle screws, allowing the exertion of powerful correction forces and strong stability. In the thoracic spine, pedicle screws are routinely placed to remove the inferior articular facet with an osteotome [22]. On the other hand, in the lumbar spine, pedicle screws are inserted after the cortical bone is removed from the entry point. Moreover, in rigid adolescent idiopathic scoliosis, spinal osteotomies, such as Ponte osteotomies [23], may be required to release the spine and allow better correction. This allows for tissue sampling as surgical waste material.

For each AIS donor, six tissue samples are recovered from surgical waste material:(1)Peripheral blood (3–5 mL).(2)Convex spinal facet.(3)Concave spinal facet.(4)Convex paravertebral muscle.(5)Concave paravertebral muscle.(6)Spinal ligament.

For each control donor, four tissue samples are recovered from surgical waste material:(1)Peripheral blood (3–5 mL).(2)Spinal facet.(3)Paravertebral muscle.(4)Spinal ligament.


*Step 3: RNA extraction, DNA extraction, histology*


All solid tissue samples are cut into small fragments (50–100 mg each) and packaged in a sterile specimen jar filled with saline solution. Tissue samples and blood are promptly sent to the laboratory for further analysis. Peripheral blood is dedicated to DNA and RNA extraction. For solid tissue samples, in addition to DNA and RNA, a small portion is used for histological analysis, which records general tissue characteristics.

Immediate, appropriate tissue storage is required to preserve nucleic acids until extraction, which can be performed with standard procedures, idoneous for subsequent DNA and RNA analyses. Solid tissue samples need pulverization before nucleic acid extraction.


*Step 4: Gene expression analysis*


The most widely used technique to assess gene expression levels is semi-quantitative real-time PCR, where mRNA levels of the gene of interest are quantified through normalization to a reference/housekeeping gene. Other gene expression techniques can be used as well.

A preliminary investigation to identify idoneous housekeeping is strongly suggested. To be suitable for normalization, housekeeping expression should remain consistent regardless of biological or experimental changes. However, different studies showed considerable variability of many traditional housekeeping genes [24,25]; therefore, idoneous references for AIS musculoskeletal tissues should be prospectively defined, since a gene consistently expressed in one cell type may vary in another.

Once the reference genes are established, RT-PCR analysis can be executed in several formats, from single tube reactions in case there are a few genes to be analyzed, to array or microarray formats, in case of numerous target genes to be tested. Primers for quantifying gene transcripts are designed to anneal every known transcript of a specific gene.


*Step 5: Data analysis*


All obtained data are meant to be merged for overall comparison and final evaluation. By comparing results in AIS compared to controls and among the different AIS tissues, significant modulation of gene expression can be appreciated only in disease or only in certain tissue types, thus suggesting the candidate tissue(s) and the candidate gene(s) for further focused exploration of the mechanisms of gene expression regulation. Moreover, correlations between gene expression and clinical parameters allow for the evaluation of a possible involvement of selected genes in disease progression. Therefore, result interpretation provides a list of putative candidates for epigenetic regulation of disease progression based on observation specifically targeted to diseased tissues, defining a list of potential biomarkers. Depending on sample size and data distribution as assessed by normality test, parametric or non-parametric tests can be used to compare gene expression data between control and disease samples or among different AIS tissues.


*Step 6: Analysis of targeted epigenetic regulation*


The last step of the workflow includes going back to the DNA to demonstrate that the observed gene expression regulation is due to epigenetic control mechanisms acting in the related genes. This can be achieved thanks to the harvest of paired samples during explant collection, which guarantees that DNA and RNA are derived from the same samples, thus ensuring that the relationship between gene expression modulation and epigenetic modifications in the corresponding genes is reliable.

The study of epigenetic control mechanisms can be performed in different ways and at different levels. Studies can be undertaken using small cell numbers and even single cells or at large scale. Total DNA extracted with standard methods can be interrogated for DNA CpG methylation status by bisulfite treatment, which converts unmethylated cytosines to uracils while leaving methylated cytosines unaffected. Methylated cytosines can then be identified by different techniques, from methylation-specific PCR (MS-PCR) to check for individual CpGs, to sequencing of bisulfite-treated DNA in specific regions, to wide-methylation analysis with microarray or next-generation sequencing in large DNA regions, entire chromosomes, or even a whole genome [26]. Methylation analysis results will be then compared between sample groups by parametric or non-parametric statistical tests, as appropriate.

## 2. Results

### 2.1. Gene Panel Identification

We identified 28 genes that, based on the literature data, may undergo epigenetic regulation during AIS progression. The list of the identified genes together with the corresponding bibliography is reported in Table 1.

### 2.2. Sample Characterization and Nucleic Acid Recovery

Samples of bone, muscle, and ligament tissues were successfully collected and histologically characterized for each enrolled donor. No macroscopic or microscopic differences were observed between AIS and controls (Figure 2).

By pulverization of nitrogen-frozen tissues (40–100 mg) followed by immediate nucleic acid extraction, appropriate amounts (from 2.9 to 15.5 μg RNA) and purity (A260/A280 ratio of 1.8–2.0) of DNA and RNA were obtained from every sample.

### 2.3. Gene Expression Analysis for Housekeeping Gene Identification

The preliminary analysis of the appropriate housekeeping genes to be used for gene expression normalization was performed in three different AIS cases and compared to one control case. Expression data are represented in Figure 3 and reported in Appendix A.

As shown in the heatmap, there is a clear separation between bone samples on one side and muscle and ligament samples on the other. Indeed, the non-supervised hierarchical clustering (Figure 3) confirms that ligament and muscle samples preferentially cluster together, separated from bone.

Housekeeping genes were selected based on two criteria: the least interindividual variability within the same tissue type and consistent expression across all tissues, in decreasing order of relevance. Furthermore, to have roughly comparable expression with the test genes, housekeeping genes were chosen to span from high to low levels of expression, allowing them to serve as an ideal reference for genes with varying expression levels.

Our analysis identified the *PPIA* (peptidylprolyl isomerase A) gene as the most stable housekeeping gene across the analyzed tissues. For array-based gene expression analyses, normalizing to more than one reference gene is suggested. We therefore included in our housekeeping gene list the *18S* (eukaryotic 18S rRNA) gene showing high expression levels and two commonly used references, namely *B2M* (beta-2-microglobulin) and/or *GAPDH* (glyceraldehyde-3-phosphate dehydrogenase). The expression levels of these four genes in the four donor samples analyzed here (three AIS and one CTR) are shown in Figure 4.

## 3. Discussion

The present study aimed at proposing a methodological approach for the study of epigenetic markers of adolescent idiopathic scoliosis progression. Recently, epigenetic regulation of gene expression received increased attention in several fields, and epigenetics became a hotly debated topic in the context of organismal development, aging, and disease [33].

Epigenetics can be defined as the set of mechanisms regulating alternative chromatin activation states in the context of the same DNA sequence. This is achieved by DNA methylation at CpG sites by histone modifications, by proteins (polycomb and trithorax), and by non-coding RNAs. All of these processes can locally remodel the chromatin structure, which in turn affects DNA accessibility and gene activity, thus modifying the cellular/organismal phenotype in a heritable yet reversible manner [33]. DNA methylation is the most studied mechanism and involves the addition of a methyl group to the five-carbon position of CpG sites. CpG methylation at regulatory regions (promoters, enhancers) influences the transcription of nearby or related genes.

Epigenetic marks are dynamic, especially during development and cell differentiation, when epigenetic programming is sensitive to environmental influences. So, the epigenome contributes to the organismal phenotype extending the genetic background and mediating the environmental inputs. The epigenetic signature is maintained through cell division; however, there is increasing evidence that epigenetic alterations can occur in terminally differentiated cells. Changes to the epigenome have been related to a series of pathological conditions including age-related diseases, cancer, diabetes, neurological disorders [33], and AIS [9,10].

Recent data have pointed to a putative role of epigenetics in AIS progression [9,10]. There are currently no reliable markers of AIS progression to guide therapeutic choices, despite the critical role they could play in the management of this condition, affecting 2–3% of the adolescent population [3].

Given the multifactorial nature of AIS and the function of epigenetics in linking genetic background to environmental influence, the availability of epigenetic progression markers may be critically important.

The genes we identified by literature search as putative candidates of AIS progression and included in our analysis panel comprise molecules playing a role in musculoskeletal tissues activity and development, transcription factors, and regulatory RNAs. The *MYH3* gene, encoding embryonic myosin and whose mutations can cause skeletal disorders, was included in the gene list since the expression of this structural molecule was found to be imbalanced in AIS convex and concave paravertebral muscles [12]. Asymmetric expression was also described for the *MSTN* gene in AIS paravertebral muscles in one of two cohorts of AIS patients [12]. This gene codes for myostatin, a protein prevalently expressed in skeletal muscle and acting as a negative regulator of muscle development and growth. *MSTN* inactivation (convex side) or overexpression (concave side) could partially account for the volumetric muscle imbalance between the two sides of the AIS curve [12]. Hyaluronan synthase 2 (HAS2), playing a critical role in disk development, also belongs to the first group. Demethylation in regulatory regions of this gene is supposed to potentially impair normal spine development, thus promoting AIS progression [16]. Other molecules of the first group are as follows: neuropeptide Y (NPY) [14,34], a fundamental regulator of bone homeostasis and osteoblasts, with a role in the response to mechanical stimuli and in osteoarthritis [14,34]; the members of the WNT/β-catenin (CTNNB1) signaling pathway, of crucial importance in proliferation, regeneration, embryonic development, and morphogenesis of various tissues including the musculoskeletal ones [14,35]. WNT/β-catenin is the canonical pathway in which the activation of β-catenin stimulates osteoblast proliferation and differentiation and prevents osteoclastogenesis as well as osteocyte and osteoblast apoptosis [36]. Canonical WNT signals are transduced through frizzled receptor and LRP5/6 coreceptor to downregulate GSK3β activity. APC participates by bringing β-catenin to GSK3B with consequent β-catenin phosphorylation and proteasomal degradation resulting in reduced bone formation and increased bone resorption [14]. A series of circulating miRNAs was found to display a peculiar expression pattern in AIS [27] and in turn regulate WNT members, which have therefore been included among those to analyze (*WNT1*, *WNT10A*, *CTNNB1*, *FRZB*, *FZD1*, *GSK3B*, *LRP5*, *LRP6*, *DKK1*, *AXIN1*, and *APC*) [14,27] together with the *FGF4* gene, involved in the development of the vertebral column and associated tissues [37] and correlated to the WNT pathway through GSK3B [30]. The *ADIPOQ* gene codes for adiponectin, a hormone involved in myotube differentiation during skeletal muscle development [38]. It demonstrated inconsistent expression in the two sides of the paravertebral muscle flanking the AIS scoliotic curve in association with disease severity and age of initiation [12].

Estrogen receptors have been extensively studied at the genetic level for possible association with AIS predisposition and severity; moreover, *ESR1* and *ESR2* methylation status was found to be associated with AIS severity [15] and occurrence [28], respectively. *FBN1* and *FBN2* genes were included in the list of potential markers due to their demonstrated role in AIS pathogenesis [39] and the reduced level of their expression found in AIS compared to control donors [29], compatible with epigenetic regulation. GREM-1 directly binds bone morphogenetic proteins (BMPs) and acts as an antagonist, thus regulating skeletal homeostasis. Given that reduced expression of GREM1 through miR-151a-3p was documented in primary osteoblasts as associated with AIS severity [31], this gene was included in the panel of putative epigenetic biomarkers. For the *PCDH10* (protocadherin 10) gene of the cadherin superfamily, which plays an important role in cell migration, there is not a described role in bone and cartilage development, even if it is involved in the WNT pathway. However, its expression was found to be inhibited in AIS osteoblasts compared to controls [40] in association with promoter hypermethylation [18]. Cartilage oligomeric matrix protein (COMP) is an extracellular protein primarily expressed in cartilage and other musculoskeletal tissues, playing an important role in bone growth. The methylation of its promoter was correlated to gene silencing and AIS progression [17].

*CRTC1* (or *TORC1*) is another gene included in the panel. It acts as a transcriptional coactivator for CREB1, thus regulating the expression of specific CREB-activated genes. It is an inducer of mitochondrial biogenesis in muscle cells [41], thus promoting muscle adaptation and skeletal muscle performance. Its expression can be regulated by methylation, as described in Alzheimer’s disease where two *CRTC1* promoter regions were found to be demethylated in the hippocampus compared to the controls [42]. Moreover, *CRTC1* expression level is correlated with MIR4300 expression and AIS curve magnitude [13]. For all these reasons, *CRTC1* appears to be a good candidate for epigenetic regulation in AIS.

Among transcription factors, the *PITX1* (pituitary homeobox 1) gene is a transcriptional regulator whose alterations are associated with many bone-related diseases. It is downregulated by promoter hypermethylation in association to AIS severity [43]. The key regulator of chondrocyte differentiation and skeletal development SOX9 transcription factor was also included in the list of genes to be investigated; *SOX9* genetic variants were associated with AIS severity [32], and upregulated expression of *SOX9* was described in AIS spinal facets [44] as induced by ghrelin hormone through the ERK/STAT3 signaling pathway. Among other functions, ghrelin participates in regulation, growth hormone secretion, bone formation, and primary chondrocyte proliferation with consequent abnormal cartilage development characteristic of AIS [44]. H19 is a long non-coding RNA promoting skeletal muscle differentiation and regeneration [45] with differential expression in the concave and convex sides of the AIS paravertebral muscle.

A large part of the literature’s evidence on the selected genes comes from studies on peripheral blood [14,16,17,18,39,43] instead of the relevant diseased tissues, mostly because they are hardly accessible, while blood is easily accessible with minimally invasive procedures. This implies the evidence obtained on peripheral blood cells needs validation in the tissues directly affected by the disease. This is a critical point since plasticity of gene expression through epigenetic regulation is time- and cell-type-dependent [33,46]; therefore, different tissues from the same organism might display different epigenetic marks and undergo different gene expression modulations. That is why a reliable analysis of epigenetic regulation in AIS cannot be performed in peripheral blood alone without considering the different tissues involved but needs a comprehensive evaluation. So, we planned a study in which to test promising candidate epigenetic AIS markers in bone, muscle, and ligament tissues positioned at the scoliotic curve. The literature data also suggested that disease onset and progression may involve other tissues, such as the intervertebral disk and vertebral body [47]. Nevertheless, we have not included these tissues in our proposed workflow, since the surgical procedure for treating AIS employs an “all posterior” approach, which does not allow direct manipulations on the intervertebral disk or vertebral body. This approach achieves excellent results in curve correction and functionality without the need to intervene on the anterior column where the spinal cord is located, thereby reducing surgical neurological risks.

For a gene to be identified as a candidate biomarker of epigenetic regulation, it is first of all necessary to verify that it experiences a variation in gene expression as the studied condition develops/progresses. This variation should occur in cells and tissues involved in the studied mechanism/disease. Moreover, the causal relationship between epigenetic expression regulation and the effect on the tissue itself, as well as its contribution to pathology development/progression, should be postulated and demonstrated based on the function of the putative gene/locus. Achieving this goal is a complex task and the proposed protocol is valuable as it allows for the exploration of gene expression in multiple different tissues of the same subject, as well as the investigation of epigenetic alterations in the corresponding DNA, obtained from the same samples from which the RNA was extracted.

On the other hand, what happens in the tissues is mirrored in the peripheral blood and is a necessary condition for a molecule to become a biomarker. Blood can be obtained from all patients, while solid tissues can only be obtained from those undergoing surgery. By collecting blood and musculoskeletal tissues (bone, muscle, and ligament) at the same time from the same AIS patient, the proposed protocol allows for the assessment of whether the expression in the last is mirrored in the peripheral blood and can therefore be used as a surrogate marker of gene expression in musculoskeletal tissues, potentially applicable to all patients by simple blood analysis.

The protocol also includes further discrimination between concave and convex bone facets and paravertebral muscles due to the high asymmetry induced by the disease. Actually, AIS risk loci associated with muscle biogenesis [48] point to a relevant role of paravertebral muscle in the onset and progression of AIS. Moreover, asymmetric expression of genes like *H19*, *ADIPOQ* [12], and *ESR1* and *2* [15,28] suggests potential difference in epigenetic regulation between the two sides of the scoliotic curve in terms of muscle tissue, while transcriptomic differences were not described for bone. We cannot rule out a similar mechanism in bone tissue, and the proposed workflow enables comparison not only between two sides of the same tissue but also between different muscle, bone, and ligament compartments, providing an unprecedented opportunity for a comprehensive analysis of gene expression modulation. Transcriptomic differences between the two sides of the AIS curve could account for structural and functional imbalance in disease progression.

The collection of healthy tissues is an onerous duty, due to possible ethical and regulatory issues specific to each individual country. In our case, the collected tissues represented surgical waste material, since the surgical technique used to treat patients with progressive AIS already involves the sampling of peripheral blood and the removal of bone, cartilage, ligamentous, and muscular tissues in both the concave and convex parts of the apex of the deformity. This ensured that no ethical or regulatory issues were present. This is especially true for control tissues, which are generally collected from the surgical waste material of patients undergoing spinal surgery for other reasons than AIS. Obtaining control tissues from healthy subjects is complicated by ethical and regulatory issues.

We demonstrated that small fragments of these tissues can be efficiently used for array-based gene expression analysis, here applied to identify housekeeping genes with consistent expression in the analyzed tissues. In the proposed protocol, we were able to obtain a sufficient amount of high-quality RNA and DNA to carry out the entire workflow for the identification of genes epigenetically regulated during AIS progression.

The first aspect of any experiment looking at relative quantitation of gene expression should be the selection of endogenous control genes, to normalize data [49]. In this proof-of-concept study, we used a commercial array consisting of 32 genes selected from published data for their constitutive and moderate abundance across numerous human tissues. Despite a clear distinction between bone on one side and ligament and muscle on the other, the *PPIA* (peptidylprolyl isomerase A) gene, a trans-isomerase enzyme involved in protein folding, demonstrated the best consistency in our experimental model and is therefore suggested as the principal reference. However, the analysis conducted here should be interpreted with respect to some limitations due to the small number of cases. We acknowledge that the characteristics and clinical histories of the three AIS patients and of the control patient may not be representative of the corresponding population. Therefore, attention should be paid before generalizing the results, and it cannot be excluded that in a wider case series, other housekeeping could display higher consistency and represent a better choice.

On the other hand, the procedure is intended for larger-scale research where a power analysis is performed to set the required number of cases based on the kind and number of genes to be investigated, to reach reliable evidence.

In this proof-of-concept study, we planned the entire workflow, we identified the gene list (including the housekeeping), and we validated experimental collection of the tissues, nucleic acids recovery, and analysis by array-based semi-quantitative PCR. The identified list of putative epigenetic biomarkers together with the consistent reference can be customized on a gene expression array to test at once all the identified putative biomarkers for higher throughput. Following the acquisition of gene expression analysis results and statistical analysis, genes exhibiting differential expression (up- or downregulation) in various tissues can be identified and their correlation with clinical data assessed. This makes it possible to compile a new, restricted list of genes that are highly likely to be epigenetically regulated during AIS development and progression. DNA from the same samples used for RNA analysis will then be interrogated for DNA methylation profiles finally demonstrating epigenetic control mechanisms in these selected genes.

The proposed workflow contains some limitations. Firstly, the feasibility needs to be verified in a larger sample size. Secondly, the use of donors with degenerative spinal disease as controls is not optimal, mostly due to a lack of age matching. However, the use of tissues from healthy controls represents a significant challenge from both an ethical and regulatory perspective. Further stratification of the AIS population into subgroups with varying progression and severity may enable comparisons among disease groups instead of controls, thus emphasizing changes associated with disease progression rather than onset.

Concerning the experimental plan, we cannot exclude the fact that altered gene expression is due to other mechanisms than epigenetic regulation. The detection of a correlation between epigenetic regulation and disease progression does not necessarily imply that epigenetic control contributes to disease progression. The workflow allows for an explorative investigation; after the identification of epigenetically regulated genes, further studies will be needed to demonstrate the causative link between these modifications and disease progression. More experimental evidence is required to show that epigenetic alterations impact the course of disease rather than being a secondary effect. Additional in vitro experiments and further patient stratification according to clinical characteristics such as skeletal maturity would probably help in identifying when epigenetic modifications occur and their putative role.

Once epigenetic control has been demonstrated, the epigenetic layout of specific genes (one or more genes) could constitute a marker of progression with predictive power on disease evolution. Epigenetic biomarkers could be used alone or, more likely, in combination with other already-used biological and/or clinical parameters to improve prognosis and treatment in a context of personalized medicine.

## 4. Materials and Methods

### 4.1. Patients and Samples

Three AIS patients (2 male and 1 female, aged 17, 23, and 14, respectively) and 1 control donor (female, aged 37) undergoing spinal surgery were enrolled in this feasibility study. Inclusion criteria for AIS patients: diagnosis of adolescent idiopathic scoliosis with progressive curve (>40° at diagnosis) necessitating surgical intervention. Exclusion criteria for AIS patients: diagnosis of scoliosis of different etiology (infective, traumatic, neoplastic, or syndromic) and/or not falling into the above criteria. Inclusion criteria for the control patient: surgery for degenerative spinal disease. No exclusion criteria were indicated for the control donor.

This study was carried out in compliance with the Helsinki declaration and approved by the local ethical committee (CE-AVEC Prot. EpigenAIS N. CE-AVEC 487/2022/Sper/IOR), including the documentation of written patient consent.

Tissue samples of spinal facets, paravertebral muscles, and spinal ligament together with peripheral blood were collected from surgical waste material of each donor.

All solid tissues samples were cut into small fragments (50–100 mg each) and packaged in a sterile specimen jar filled with saline solution and then immediately sent to the laboratory along with a sterile tube containing EDTA non-coagulated blood.

Solid tissue samples were cut into three parts, weighted, snap frozen, and divided as follows: one part submerged in RNALater (Invitrogen, Thermo Fisher Scientific, Paisley, United Kingdom) and stored at −20 °C until RNA extraction, one part nitrogen-frozen until DNA extraction, and the third part processed for histology.

Peripheral blood was partly dedicated to RNA extraction (2–4 mL): after red blood cell lysis through 2 × 10 min incubation in 150 mM NH4Cl, 10 mM KHCO_3_, and 0.1 mM EDTA, pH 7.2–7.4, pelleted cells were submerged in RNALater and stored at −20 °C until RNA extraction. The remaining blood (1 mL) was stored at −20 °C for DNA extraction.

### 4.2. Total RNA Extraction

Tissue samples stored in RNALater were returned to room temperature and then retrieved with sterile forceps. Excess liquid was blotted away, and samples were liquid-nitrogen-frozen in 5 mL PFTE flasks containing a stainless-steel grinding ball and then pulverized with the Mikro-Dismembrator S grinding mill (Sartorius Stedim Italy SpA, Varedo, Italy) at 2000 rpm for 45″. Regarding peripheral blood, RNALater was carefully removed.

Total cellular RNA was extracted from homogenized tissues and from blood using the Trifast isolation reagent (VWR, Milan, Italy) following manufacturer’s instructions. Total RNA was eluted in RNase free water, spectrophotometrically quantified, and then stored at −80 °C.

### 4.3. Total DNA Extraction

Tissue samples stored in liquid nitrogen were pulverized as described in the previous paragraph while blood samples were returned to room temperature. DNA was extracted using the PureLink Genomic DNA Mini Kit (Invitrogen, Carlsbad, CA, USA), eluted in DNase free distilled water, quantified by spectrophotometric determination, and then stored at −80 °C.

### 4.4. Histology

One fragment of each tissue sample was formalin-fixed and paraffin-embedded following standard procedures. Sections of 5 µm were cut and stained with hematoxylin-eosin (Bioptica, Milan, Italy), and representative images were recorded.

### 4.5. Gene Expression Analysis

Array-based semi-quantitative real-time RT-PCR was used to identify the idoneous housekeeping genes for the tissues analyzed.

For cDNA synthesis, 1 μg of RNA was reverse transcribed by random priming with SuperScript Vilo MasterMix (Invitrogen, USA) following manufacturer’s instructions, and then cDNA samples were tested by the Taqman Array Human Endogenous Control (ThermoFisher Scientific, Waltham, MA, USA) containing 32 housekeeping candidates. About 10 ng of cDNA per sample was amplified in triplicate in a 20 μL reaction volume in presence of Taqman Universal Master Mix II, no UNG (ThermoFisher Scientific, USA). Reaction conditions: 10′ at 95 °C followed by 40 cycles (15″ at 95 °C, 60″ at 60 °C). Results were visualized by QuantStudio Design and Analysis Software v. 1.5.2 (ThermoFisher Scientific, USA) and by the RT2 Profiler PCR Array data analysis tool (Qiagen, Hilden, Germany). Genes with the higher consistency across tissues were evaluated by comparing cycle threshold (Ct). Relative expression was obtained by normalization to one of the 32 putative housekeeping analyzed.

To evaluate results, mean relative expression from the 3 AIS cases was used.

## 5. Conclusions

In conclusion, we have proposed here an original approach to the study of epigenetic regulation of AIS progression, allowing for the successful identification of efficient epigenetic biomarkers. The approach is based on a detailed preliminary investigation on tissues directly involved in disease development and on the actual modulation of target gene expression. Differential tissues from AIS patients were used to assess the local gene expression of a series of genes with potential roles as epigenetic biomarkers of AIS progression. The combination of gene expression and genetic information from the same tissue samples may provide a significant added value to improve our understanding of epigenetic control of AIS. The proposed approach can be applied to different experimental and disease models or to different tissues and experimental plans and therefore could become a flexible and suitable tool for conducting epigenetic studies.

The feasibility of the proposed methodological protocol was originally tested here concerning the identification of the putative epigenetic markers of AIS progression, the collection of the different tissues, the retrieval of the appropriate amount and quality of RNA and DNA, the identification of suitable housekeeping reference genes idoneous to the analyzed tissues, and the design of a gene expression array to test at once all the identified putative biomarkers.

## Figures and Tables

**Figure 1 ijms-25-05329-f001:**
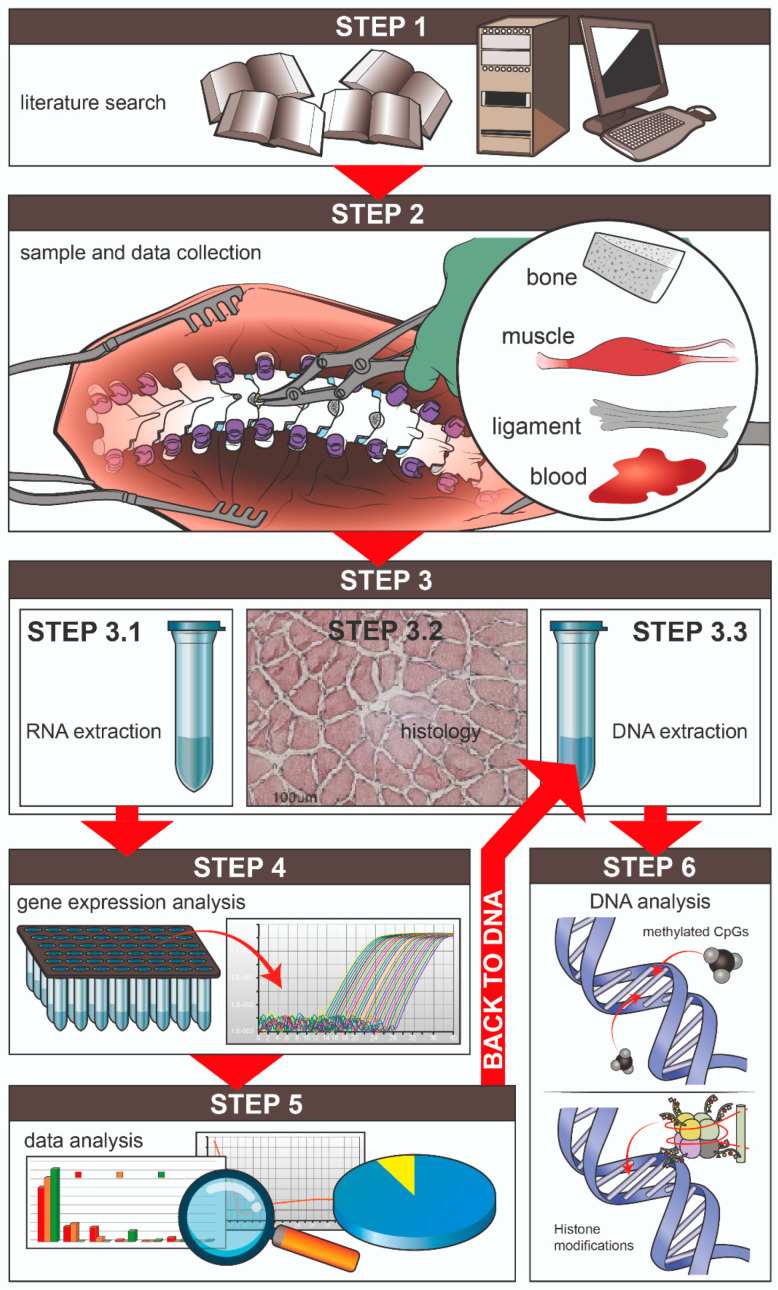
Workflow to assess local gene expression of putative epigenetic regulators of AIS progression. Step 1: literature search, identification of the genes of interest, and setup of the gene panel; Step 2: collection of paravertebral tissues (muscle, bone, and ligament) and peripheral blood from donors undergoing spine surgery; Step 3: total RNA extraction, total DNA extraction, and histological characterization of each sample; Step 4: array-based gene expression analysis; Step 5: data analysis and identification of suggestive markers; Step 6: back to DNA to confirm epigenetic regulation (CpGs methylation/histone modifications).

**Figure 2 ijms-25-05329-f002:**
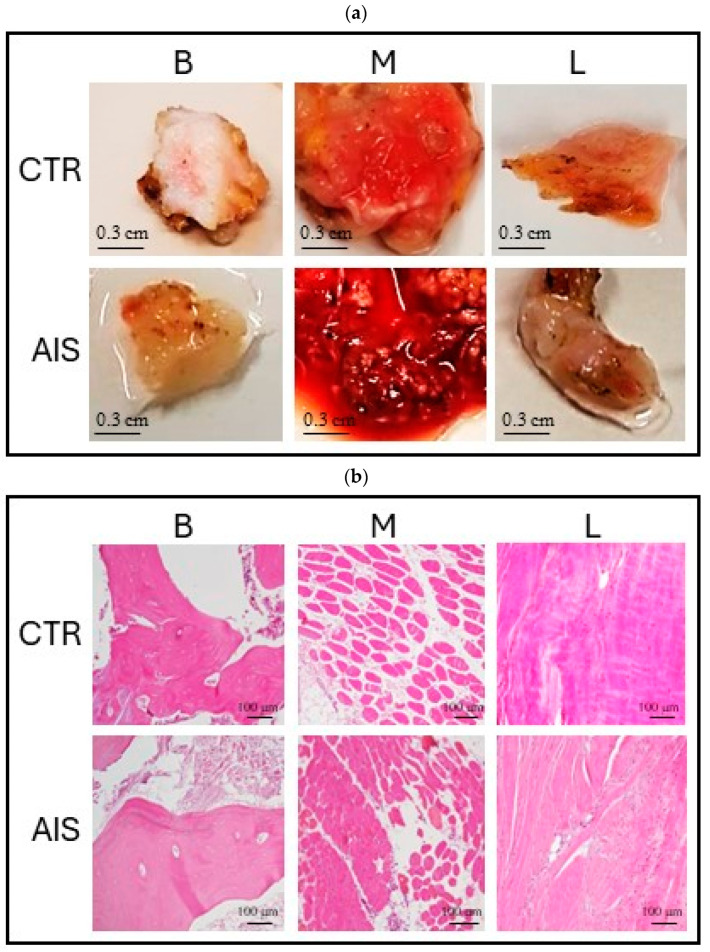
Bone, muscle, and ligament tissue samples. (**a**) Macroscopic images of bone, muscle, and ligament samples from the control donor (top) and a representative AIS donor (bottom); (**b**) hematoxylin–eosin staining of bone, muscle, and ligament samples from the control donor (top) and a representative AIS donor (bottom). B = bone (spinal facet); M = paravertebral muscle; L = spinal ligament; CTR = control; AIS = adolescent idiopathic scoliosis.

**Figure 3 ijms-25-05329-f003:**
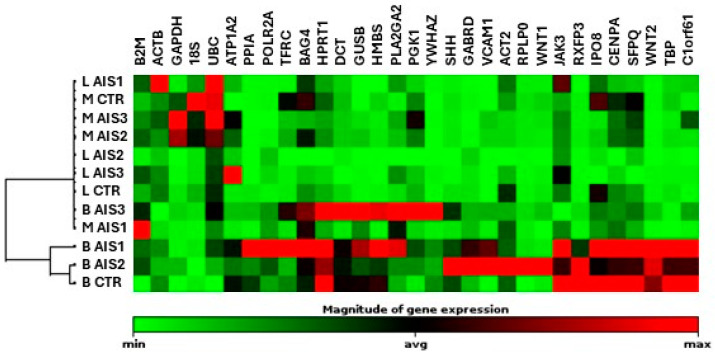
Gene expression analysis of a panel of putative housekeeping genes in bone, muscle, and ligament tissues. The dendrogram was obtained by hierarchical clustering of bone facet, paravertebral muscle, and spinal ligament of 3 AIS donors and 1 control (CTR) donor based on the expression of 32 putative housekeeping genes. Gene expression levels are shown as a heatmap for each gene, with red indicating high relative expression and green low relative expression (data are visualized by using one of the 32 genes, namely *CD58*, as reference for normalization). AIS = adolescent idiopathic scoliosis; CTR = control; B = bone facet; M = paravertebral muscle; L = spinal ligament.

**Figure 4 ijms-25-05329-f004:**
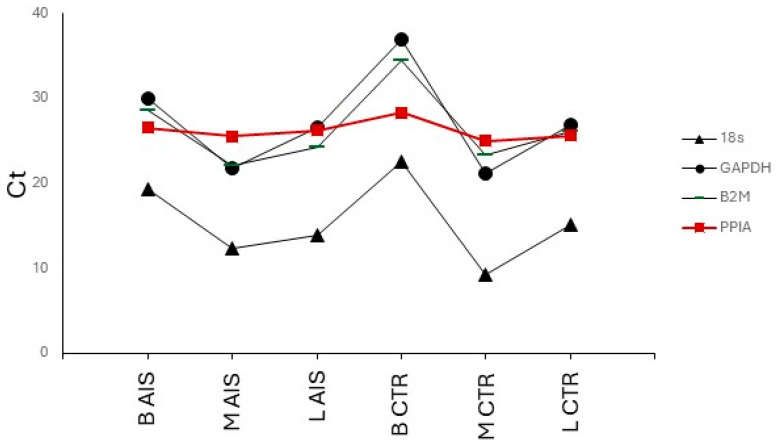
*PPIA*, *18S*, *GAPDH*, and *B2M* housekeeping gene expression across the analyzed tissues. Mean expression levels are shown as Ct (threshold cycles). AIS = adolescent idiopathic scoliosis; CTR = control; B = bone facet; M = paravertebral muscle; L = spinal ligament.

**Table 1 ijms-25-05329-t001:** List of the selected candidate epigenetic regulators of AIS progression.

Gene Symbol	Gene Name	Ref.
*ADIPOQ-*	Adiponectin, C1Q, and collagen domain containing	[12]
*APC-*	APC, WNT signaling pathway regulator	[27]
*AXIN1-*	Axin 1	[14]
*COMP-*	Cartilage oligomeric matrix protein	[17]
*CRTC1-*	CREB-regulated transcription coactivator 1	[13]
*CTNNB1-*	catenin beta 1	[14]
*DKK1-*	Dickkopf WNT signaling pathway inhibitor 1	[14]
*ESR1-*	Estrogen receptor 1	[15]
*ESR2-*	Estrogen receptor 2	[28]
*FBN1-*	Fibrillin 1	[29]
*FBN2-*	Fibrillin 2	[29]
*FGF4-*	Fibroblast growth factor 4	[30]
*FRZB-*	Frizzled-related protein	[14]
*FZD1-*	Frizzled class receptor 1	[14]
*GREM1-*	Gremlin 1, DAN family BMP antagonist	[31]
*GSK3B-*	Glycogen synthase kinase 3 beta	[14]
*H19-*	Imprinted maternally expressed transcript (non-protein coding)	[12]
*HAS2-*	Hyaluronan synthase 2	[16]
*LRP5-*	LDL receptor related protein 5	[14]
*LRP6-*	LDL receptor related protein 6	[14]
*MSTN-*	Myostatin	[12]
*MYH3-*	Myosin, heavy chain 3, skeletal muscle, embryonic	[12]
*NPY-*	Neuropeptide Y	[14]
*PCDH10-*	Protocadherin 10	[18]
*PITX1-*	Paired-like homeodomain 1	[18]
*SOX9-*	SRY-box 9	[32]
*WNT10A-*	Wnt family member 10A	[14]
*WNT1-*	Wnt family member 1	[14]

## Data Availability

The original contributions presented in this study are included in the article/Appendix A; further inquiries can be directed to the corresponding author.

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
