# Peer review of "Identification of Epigenetic Biomarkers of Adolescent Idiopathic Scoliosis Progression: A Workflow to Assess Local Gene Expression"

_ijms, 2024, doi:10.3390/ijms25105329_

Round 1

Reviewer 1 Report

Comments and Suggestions for Authors

Dear Authors,

Thank you very much for submitting your research to IJMS.

Based on the detailed review of the manuscript "Identification of epigenetic biomarkers of Adolescent Idiopathic Scoliosis progression: a workflow to assess local gene expression", here are my comments for the authors

1. The study's focus on epigenetic markers in local tissue samples rather than peripheral blood represents a novel approach that could offer more direct insights into AIS progression mechanisms.

2. The authors have structured a comprehensive and detailed methodology to study gene expression and epigenetic regulation in tissues directly involved in AIS, providing a significant contribution to the field of scoliosis research. Their work, supported by AO Foundation AO Spine, follows a rigorous experimental protocol, from patient sample collection to data analysis, and includes a thorough review of the literature to identify candidate genes for epigenetic regulation in AIS progression.

3. The detailed description of the experimental workflow, from the literature review for gene panel identification to the analysis of gene expression and epigenetic regulation, provides a robust framework for the study.

4. By aiming to identify reliable biomarkers for AIS progression, the research has significant potential clinical implications, offering new avenues for diagnosis and the development of targeted therapies.

5. The study involves a relatively small and potentially non-representative sample size, which may limit the generalizability of the findings. Expanding the study to include a larger and more diverse cohort would strengthen the evidence for the identified epigenetic markers.

6. The detailed description of the experimental workflow, from the literature review for gene panel identification to the analysis of gene expression and epigenetic regulation, provides a robust framework for the study.

7. By aiming to identify reliable biomarkers for AIS progression, the research has significant potential clinical implications, offering new avenues for diagnosis and the development of targeted therapies.

8. The study involves a relatively small and potentially non-representative sample size, which may limit the generalizability of the findings. Expanding the study to include a larger and more diverse cohort would strengthen the evidence for the identified epigenetic markers.

9. The control group criteria could be broadened to include individuals without spinal deformities, which might provide a clearer contrast to the AIS group and enhance the validity of the findings.

10. While the manuscript details the methods for data collection and preliminary analysis, a more in-depth explanation of the statistical methods used for analyzing differential gene expression and methylation patterns would add clarity and reproducibility to the study.

11. The manuscript would benefit from a section discussing the limitations of the current study and outlining specific future research directions, including potential clinical applications of the findings and how these biomarkers could be integrated into AIS management strategies.

12. Please try to read the followings and embed the followings if applicable.

References

(1)Investigating the Differential Circulating microRNA Expression in Adolescent Females with Severe Idiopathic Scoliosis: A Proof-of-Concept Observational Clinical Study.

Raimondi L, De Luca A, Gallo A, Perna F, Cuscino N, Cordaro A, Costa V, Bellavia D, Faldini C, Scilabra SD, Giavaresi G, Toscano A.

Int J Mol Sci. 2024 Jan 1;25(1):570. doi: 10.3390/ijms25010570.

PMID: 38203740

https://pubmed.ncbi.nlm.nih.gov/38203740/

(2) Evaluation of the regenerative capacity of stem cells combined with bone graft material and collagen matrix using a rabbit calvarial defect model.

Park JB, Kim I, Lee W, Kim H.J Periodontal Implant Sci. 2023 Dec;53(6):467-477. doi: 10.5051/jpis.2204880244. Epub 2023 Mar 28.

https://pubmed.ncbi.nlm.nih.gov/37154108/

Thank you very much.

Comments on the Quality of English Language

Dear Authors,

Please consider using English editing service for improvements.

Thank you very much.

Author Response

 We thank the reviewer for the insightful comments provided. Please, find below a detailed response; the manuscript has been amended and revised accordingly.

 Q1- The study involves a relatively small and potentially non-representative sample size, which may limit the generalizability of the findings. Expanding the study to include a larger and more diverse cohort would strengthen the evidence for the identified epigenetic markers.

A1- In this proof of concept study, we developed and presented a workflow for the study of candidate epigenetic biomarkers of AIS progression, nevertheless we did not apply the workflow to a donor case series. The putative epigenetic markers have been selected by revising the literature and not identified by experimental analyses, which can be carried out by applying the protocol presented here to confirm their involvement in the tissues of interest.

Here we only detailed the steps of the experimental approach and, in a small sample size, evaluated the accuracy of nucleic acid recovery from different tissue samples, its application to array-based gene expression analysis, and its capacity to identify reliable housekeeping genes to provide a feasibility proof.

We completely agree that results concerning housekeeping genes may not be reliable due to the very small sample size and declared this limitation in the discussion (lines 447-53). We implemented this part by adding a sentence on the sample size required for a complete study providing reliable and generalized results (lines 454-6).

The proposed experimental model can be applied also to other pathologies in which an epigenetic control can be supposed. The number of analyzed cases must be chosen each time depending on the availability of donor samples, of tissues (depending on the type of surgical procedure) and on the genes to be analyzed.

Q2- The control group criteria could be broadened to include individuals without spinal deformities, which might provide a clearer contrast to the AIS group and enhance the validity of the findings.

A2- Once it has been decided to use the proposed experimental workflow, the study groups can be selected in various ways, depending on the specific objective, and considering ethical and regulatory aspects specific of each country (as stated at lines 426-31). One can give more focus on the comparison between different tissues of the same AIS subject or on comparing AIS donor tissues with healthy control tissues. Our focus was to primarily compare epigenetic changes among vertebral tissues, possibly influencing disease progression, and controls were only included to have a referenced expression of the genes of interest.  The inclusion of tissue donors without spinal deformities would certainly help identify genes involved in disease onset, while comparing patients with different degrees of pathology could shed light on genes involved in disease progression. The extension of the proposed workflow to individuals without spinal deformities and the recruitment of healthy spinal tissues would certainly improve the knowledge about the role of putative epigenetic biomarkers. However, this would raise important ethic concerns. We added a comment on this in the workflow description (lines 120-3) and in the discussion (lines 431-4).

Q3. While the manuscript details the methods for data collection and preliminary analysis, a more in-depth explanation of the statistical methods used for analyzing differential gene expression and methylation patterns would add clarity and reproducibility to the study.

A3- In this feasibility study we did not use statistical analysis due to the very small sample size (3 AIS and 1 control donor). Descriptive statistics was solely employed to mediate housekeeping gene expression levels of the 3 AIS cases. This was added to materials and methods (line 556).

Instead, in a putative research study applying the proposed workflow, statistical analysis will be employed to analyze gene expression data. The type of test will depend on sample size and normality data distribution: parametric tests for normal data distribution (t-test for single or Anova for multiple comparisons) or non-parametric tests for non-normal data distribution (Mann-Witney/Wilcoxon for single or Kruskal-Wallis/Friedman for multiple comparisons). Similarly, about methylation pattern analysis, the data will be analyzed according to the type of experimental plan, such as purely qualitative data (presence/absence of methyllation in specific loci), or quantitative data (% of methylated cytosines in specific DNA regions or general methylation status). The above-mentioned statistical test will be then applied to compare methylation status in the different experimental groups. The “Step 5: Data analysis” paragraph and the “Step 6: analysis of targeted epigenetic regulation” paragraph were implemented with this information (lines 182-4, and 201-3, respectively).

Q4. The manuscript would benefit from a section discussing the limitations of the current study and outlining specific future research directions, including potential clinical applications of the findings and how these biomarkers could be integrated into AIS management strategies.

A4- Thank you for the suggestion. We implemented the present version of the manuscript with a paragraph commenting limitations and possible clinical applications at the end of the discussion section (lines 469-91).

Q5. Please try to read the followings and embed the followings if applicable.

References

  • Investigating the Differential Circulating microRNA Expression in Adolescent Females with Severe Idiopathic Scoliosis: A Proof-of-Concept Observational Clinical Study.

Raimondi L, De Luca A, Gallo A, Perna F, Cuscino N, Cordaro A, Costa V, Bellavia D, Faldini C, Scilabra SD, Giavaresi G, Toscano A. Int J Mol Sci. 2024 Jan 1;25(1):570. doi: 10.3390/ijms25010570. PMID: 38203740 https://pubmed.ncbi.nlm.nih.gov/38203740/

(2) Evaluation of the regenerative capacity of stem cells combined with bone graft material and collagen matrix using a rabbit calvarial defect model. Park JB, Kim I, Lee W, Kim H.J Periodontal Implant Sci. 2023 Dec;53(6):467-477. doi: 10.5051/jpis.2204880244. Epub 2023 Mar 28. https://pubmed.ncbi.nlm.nih.gov/37154108/

A5- In the introduction section, we added a note about circulating extracellular vesicles from severe AIS females containing miR-30 family members and possibly contributing to disease pathogenesis and severity (lines 83-7). We also added the corresponding reference (PMID: 38203740; ref.21).

Q6. Please consider using English editing service for improvements.

A6. A thorough review of English has been done.

Reviewer 2 Report

Comments and Suggestions for Authors

This is an interesting proposal, aiming to correlate detected blood-based epigenetic changes with tissue from AIS surgery. The hypothesis is that blood-based epigenetic markers in AIS must exert their effects on tissues involved in AIS progression (bone, ligament, muscle), and thus correlation of tissue and blood expression could yield more true markers.

Of course, this would imply that the identified genes are all affecting one of these three tissues. I think Table 1 should have a column suggesting which of the three tissues are involved. It could also be that the changes involve tissues not proposed to be sampled (especially intervertebral disc). There is a body of literature implicating the disc as an initiating event in AIS progression - could there be convenience IVD sampling in your protocol, especially if Ponte osteotomies are performed? What about the vertebral body?

The authors spend some time on an important topic - housekeeping gene investigations to ensure stable expression. How did the authors select the appropriate housekeeping gene - meaning, how were the statistics computed?   

In addition, there is no description of the proposed gene primers. Has there been thought given to selecting primers that are insensitive to any gene variants (alleles)? 

Lastly, and importantly, there should be some discussion on cause versus effect. Perhaps the observed epigenetic differences are not causative, but rather a downstream effect of the evolving deformity. Therefore, perhaps focusing on young patients with less skeletal maturity may be a better, more strict group of patients to study.

Obtaining control tissue in adolescents is surely to be very difficult, but not impossible. Older control specimens will be more difficult to compare v younger patient specimens.

Author Response

We thank the reviewer for the insightful comments provided. Please, find below a detailed response; the manuscript has been amended and revised accordingly.

 Q1- This is an interesting proposal, aiming to correlate detected blood-based epigenetic changes with tissue from AIS surgery. The hypothesis is that blood-based epigenetic markers in AIS must exert their effects on tissues involved in AIS progression (bone, ligament, muscle), and thus correlation of tissue and blood expression could yield more true markers. Of course, this would imply that the identified genes are all affecting one of these three tissues. I think Table 1 should have a column suggesting which of the three tissues are involved. It could also be that the changes involve tissues not proposed to be sampled (especially intervertebral disc). There is a body of literature implicating the disc as an initiating event in AIS progression - could there be convenience IVD sampling in your protocol, especially if Ponte osteotomies are performed? What about the vertebral body?

A1- Our hypothesis is to identify primarily epigenetic changes of specific genes in vertebral tissues that may influence the progression of AIS. Furthermore, if the presence of epigenetic changes is also mirrored in peripheral blood, the changes identified could also serve as possible biomarkers and be used to predict the progression of the pathology as a easily usable surrogate marker by simple blood analysis. We cannot assume a priori that one musculoskeletal tissue is most affected by the epigenetic regulation of a specific gene. We can infer, at most, that specific genes could play a predominant role in specific tissues based on the role they play in physiological conditions, and these aspects are briefly summarized in the discussion. However, we cannot predict whether these genes are epigenetically regulated in one, two, three, or even none of the musculoskeletal tissues studied and only the results of a larger sample size will elucidate this point.

As concerning the analysis of additional tissues, indeed recent literature suggests that the intervertebral disc and vertebral body could be the initial sites for the onset of scoliosis and disease progression (a reference has been added: PMID 35818045, ref.49). However, the surgical procedure for treating this condition employs an "all posterior" approach, which does not allow direct manipulations on intervertebral disc or vertebral body. These components, part of the anterior column, are difficult to access due to the presence of the spinal cord, which cannot be manipulated without incurring significant iatrogenic neurological risks. Moreover, by exclusively addressing the posterior structures of the spine, the "all posterior" technique achieves excellent results in curve correction and functionality without the need to intervene on the anterior column, thereby reducing surgical neurological risks. This is why tissues from the intervertebral disc or vertebral body were not included, although, theoretically, it would have made sense to do so.

We added these comments in the discussion (lines 387-94).

Q2- The authors spend some time on an important topic - housekeeping gene investigations to ensure stable expression. How did the authors select the appropriate housekeeping gene - meaning, how were the statistics computed?

A2- Housekeeping genes were selected based on a panel of commonly used housekeeping in an array commercial format (Taqman Array Human Endogenous Control, ThermoFisher Scientific, USA) whose expression was tested analyzing all types of tissue (bone, muscle, ligament, blood) from 3 AIS patients and one control. As indicated in the text we choose housekeeping genes that had consistent expression across all tissues and the least interindividual variability within the same tissue type (lines 245-7, 553-4). Also, housekeeping genes were chosen to span from high to low levels of expression allowing them to serve as an ideal reference for genes with varying expression levels, since we can't know in advance what the level of expression of the genes we're going to evaluate will be (lines 247-50). Descriptive statistics (mean) was used to summarize results from the 3 AIS cases (added at line 556). Threshold cycles were compared without statistical analysis due to the very small sample size.

Q3- In addition, there is no description of the proposed gene primers. Has there been thought given to selecting primers that are insensitive to any gene variants (alleles)? 

A3- The primers used in this feasibility study were only those of the housekeeping genes. They were included in a patented commercial array (Taqman Array Human Endogenous Control, ThermoFisher Scientific, USA). Primers and probes are designed to anneal and detect all described transcripts.

As concerning the proposed gene list (Table 1 of the manuscript), the levels of expression of all or at least as many known transcripts as possible of each gene should be evaluated, therefore primers would be designed in the common regions. This was specified in Step 4 of the workflow (lines 169-70).

Q4- Lastly, and importantly, there should be some discussion on cause versus effect. Perhaps the observed epigenetic differences are not causative, but rather a downstream effect of the evolving deformity. Therefore, perhaps focusing on young patients with less skeletal maturity may be a better, more strict group of patients to study.

A4- Thanks for this insightful comment. We fully agree the observation of epigenetic alterations does not necessarily imply that these alterations are responsible for disease progression. The proposed workflow merely represents a starting point for detecting a possible epigenetic control. To establish the causal relationship, in vitro experiments, and more selected definition of the study population such as, as suggested, focusing on a restricted population with greater skeletal immaturity, will be necessary. These considerations have been included in the Discussion section (lines 477-86).

Q5- Obtaining control tissue in adolescents is surely to be very difficult, but not impossible. Older control specimens will be more difficult to compare v younger patient specimens.

A5: Again, we fully agree that age matching between disease and control samples is relevant, but it is really challenging to obtain healthy control tissues, both for ethical and regulatory reasons. The stratification of disease samples according to disease severity could serve as a potential alternative since our study aims to detect epigenetic factors related to disease progression. This would allow different disease groups to be compared instead of disease and controls, emphasizing changes associated with disease progression rather than onset. A paragraph has been added to the Discussion explaining the limits of the presented study along with the above comments (lines 471-5).

Round 2

Reviewer 1 Report

Comments and Suggestions for Authors

Dear Authors,

Thank you very much for preparing the manuscript.